# Microstructure and Properties of NiCoCrAlTi High Entropy Alloy Prepared Using MA-SPS Technique

**DOI:** 10.3390/ma16052082

**Published:** 2023-03-03

**Authors:** Zhipei Chen, Xiaona Ren, Peng Wang, Jiangxiong Hu, Changchun Ge

**Affiliations:** 1Institute of Powder Metallurgy and Advanced Ceramics, University of Science & Technology Beijing, Beijing 100083, China; 2China Machinery Institute of Advanced Materials (Zhengzhou) Co., Ltd., Zhengzhou 450001, China

**Keywords:** high entropy alloy, mechanical alloying, spark plasma sintering, composition design, superalloy

## Abstract

In this study, Ni_35_Co_35_Cr_12.6_Al_7.5_Ti_5_Mo_1.68_W_1.39_Nb_0.95_Ta_0.47_ high entropy alloy (HEA) was prepared using mechanical alloying (MA) and spark plasma sintering (SPS) based on the unique design concept of HEAs and third-generation powder superalloys. The HEA phase formation rules of the alloy system were predicted but need to be verified empirically. The microstructure and phase structure of the HEA powder were investigated at different milling times and speeds, with different process control agents, and with an HEA block sintered at different temperatures. The milling time and speed do not affect the alloying process of the powder and increasing the milling speed reduces the powder particle size. After 50 h of milling with ethanol as PCA, the powder has a dual-phase FCC+BCC structure, and stearic acid as PCA inhibits the powder alloying. When the SPS temperature reaches 950 °C, the HEA transitions from a dual-phase to a single FCC phase structure and, with increasing temperature, the mechanical properties of the alloy gradually improve. When the temperature reaches 1150 °C, the HEA has a density of 7.92 g cm^−3^, a relative density of 98.7%, and a hardness of 1050 HV. The fracture mechanism is one with a typical cleavage, a brittle fracture with a maximum compressive strength of 2363 MPa and no yield point.

## 1. Introduction

The aviation industry has thrived with global integration and technological progress [1]. The “heart” of an aircraft is the aero-engine, which, after more than a half-century, has reached a fourth-generation design, with the fifth currently in development. Aero-engines work in extreme temperature, rotation-speed, and stress environments, so material performance and safety requirements are stringent. This is especially true for the turbine disk, whose design and material characteristics are paramount for aero-engine performance, and a significant research and development challenge. As a result, the aerospace industry desperately needs alloy materials that can withstand high temperatures [2]. Powder superalloys are well suited for such critical components [3,4,5] and fourth-generation powder superalloy turbine disks are in development to withstand temperatures reaching 815 °C. As temperature and performance requirements rise, new design concepts, compositions and processes must be developed for a new generation of powder superalloy turbine disks [6,7,8] and to mitigate the length and high cost of new material development. High entropy alloys (HEAs) with novel designs are an increasing focus of research, and related concepts are gaining in popularity. HEAs are alloys with at least five primary elements, each with an atomic percentage between 5 and 35%, and any minor element less than 5% [9]. While HEA has outstanding properties, including high strength, hardness, high-temperature corrosion resistance, and irradiation resistance [10,11], research on HEAs is still limited. The unique design concept of HEAs can be applied to the innovation of a new generation of powder superalloys and the development of turbine disk components suitable for the extreme working conditions of the aero-engine.

Combining MA (mechanical alloying) with SPS (spark plasma sintering) to prepare HEAs has recently received much attention. MA overcomes phase diagram limitations by using mechanical energy to effect the atomic alloying of solid components, producing alloy systems inaccessible to conventional procedures and increasing powder homogeneity and solid solubility. SPS is a new rapid sintering technology commonly used to prepare high-performance HEA blocks. It combines plasma activation and hot-pressing sintering and benefits from fast heating and cooling rates, short sintering times, controllable microstructures, energy savings, and low environmental impact. Pulse current heating and high pressure can rapidly heat and sinter the powder particles into sintered blocks [12,13,14]. However, many challenges remain in preparing HEAs using MA and SPS, such as the selection of initial elements, setting of milling parameters, selecting process control agents (PCAs), and controlling the sintering temperature.

In this study, the HEA composition design was based on the third-generation powder superalloy FGH100L, applying the criteria of the HEA phase formation rules. The combination of MA and SPS successfully produced the Ni_35_Co_35_Cr_12.6_Al_7.5_Ti_5_Mo_1.68_W_1.39_Nb_0.95_Ta_0.47_ HEA. The effects of different milling parameters and sintering temperature on the powder’s phase structure and microstructure, and the HEA block properties, were investigated and optimal conditions were applied to prepare high-performance HEA blocks. The results provide a reference for the development of a new generation of powder superalloy turbine disks. 

## 2. Experimental Procedure

Elemental powders of Ni, Co, Cr, Al, Ti, Mo, W, Nb, and Ta with a purity of >99.95% and particle size of ≤50 μm were used as raw materials. Powders were milled in a four-tank planetary ball mill (1L volume, 60% filled coefficient of balls and powders) with a ball-to-powder ratio of 10:1, at speeds of 300 and 500 r min^−1^ for up to 80 h. To avoid contamination with other elements, the milling balls and ball milling tank materials were Fe-Cr stainless steel. To improve the degree of powder alloying, four kinds of balls with diameters of 5, 8, 12, and 15 mm were used. Ethanol and stearic acid were used as PCA (2% of the total weight of the balls and powder) to prevent powder agglomeration during ball milling. Powders were weighed and charged in a glove box filled with argon. Powder microstructure and phase structure were analyzed at 15 h ball milling intervals. To avoid changing the ball-to-powder ratio, after sampling the powder 2–3 times, the ball mill tank and balls were cleaned, and the powder was re-weighed in the glove box.

For SPS sintering, the milled powders were poured into a graphite mold (15 mm diameter), the sintering temperature ranged from 950–1150 °C, the heating rate was 100 °C min^−1^, and the pressure was kept constant at 50 MPa throughout the operation. After reaching the desired sintering temperature, the holding period was 10 min and, after sintering, the samples were cooled to room temperature in a vacuum.

Phase structures of the powders and sintered samples were investigated using X-ray diffraction (XRD, Smartlab, Rigaku, Tokyo, Japan). Various analysis methods were applied to characterize the powders and sintered samples, including field-emission scanning electron microscopy (FESEM, Regulus8100, Hitachi, Nagano, Japan) and energy-dispersive X-ray spectroscopy (EDS, Esprit Compact, Bruker, Billerica, MA, USA). A Diamond differential scanning calorimeter (DSC, Perkin-Elmer, Waltham, MA, USA) was used for the thermal analysis of powders, with a heating rate of 10 °C min^−1^ and maximum temperatures of 1250 and 1300 °C. The real density of the sample was measured using the drainage method. A Micro Vickers hardness tester (MH-6, Shanghai HengYi precision instrument Co., Ltd., Shanghai, China) was used to test the hardness of the sample, applying a pressure load of 500 gf (4.9 N) and a holding time of 20 s. Eight random measurements were made on the sample surface and, after removing the maximum and minimum values, the average value was taken as the final result. The oxygen content of the powder was measured using LECO OHN8 and the carbon content using LECO CSO844 (LECO Corporation, San Jose, CA, USA). A universal testing machine was used to test the quasi-static compressive strength of the samples (UTC 2017-042, MTS, Eden Prairie, MN, USA), for which sintered samples were cut into cylindrical samples (ϕ 3 mm × 6 mm) and polished. Unidirectional axial pressure was applied to the sample at a stress rate of 1 × 10^−3^ s^−1^.

## 3. Result and Discussion

### 3.1. Prediction of Phase Formation

Since HEAs are composed of at least five elements, it is difficult to predict the composition and structure of alloys via phase diagram calculations and first principles. Therefore, new research methods need to be developed [15]. Zhang et al. proposed the widely-recognized HEA solid-solution theory based on the rules of alloy phase formation [16,17]. According to the rules of phase formation, an HEA solid solution can be formed as long as the parameters are within a theoretical range when designing the HEA composition. The main parameters ranges are atomic radius difference (0 ≤ δ ≤ 6.6), mixing enthalpy (−15 ≤ ∆H_mix_ ≤ 5 kJ mol^−1^), mixing entropy (11 ≤ ∆S_mix_ ≤ 19.5 kJ mol^−1^), and thermodynamic constant (Ω > 1.1). In this study, based on FGH100L and the HEA design concept, the HEA composition was prepared with Ni, Co, Cr, Al, and Ti as the main elements, and Mo, W, Nb, and Ta as secondary elements. The main components of the alloy were gradually adjusted according to the principle of entropy increase, as shown in Figure 1. Four kinds of HEA were designed (Ⅰ–Ⅳ): Ni_35_Co_35_Cr_12.6_Al_7.5_Ti_5_Mo_1.68_W_1.39_Nb_0.95_Ta_0.47_;Ni_27.55_Co_27.55_Cr_27.55_Al_7.5_Ti_5_Mo_1.68_W_1.39_Nb_0.95_Ta_0.47_;Ni_22.54_Co_22.54_Cr_22.54_Al_22.54_Ti_5_Mo_1.68_W_1.39_Nb_0.95_Ta_0.47_; andNi_19.03_Co_19.03_Cr_19.03_Al_19.03_Ti_19.03_Mo_1.68_W_1.39_Nb_0.95_Ta_0.47_.

The δ, ∆H_mix_, ∆S_mix_, Ω, and other factors were calculated for the HEAs with the above four compositions. Table 1 shows the nominal composition of FGH100L. Table 2 shows the alloy elements’ crystal structure, molar mass, and melting point. 

Atomic size differences determine the phase structure of alloys. Hume-Rothery pointed out that atomic size factors greatly influence the solid solubility of binary alloys. The more significant the size difference between atoms, the lower the solid solubility, and the greater the degree of lattice distortion after the atoms are dissolved into the lattice. At the same time, the diffusion rate between the principal elements will be reduced, resulting in a lower phase transition rate [18,19]. Based on the solid-solution theory of binary alloys, the average atomic radius of each principal element is substituted into Equation (1) to determine the δ of each alloy system:(1)$$\delta =100\sqrt{\sum_{i=1}^{n}c_{i}{\left(1-\frac{r_{i}}{\;\bar{r}}\right)}^{2}}\;\bar{r}=\sum_{i=1}^{n}c_{i}r_{i}$$
where c_i_ and r_i_ are the mole percentage and atomic radius of principal element i, respectively. ∆H_mix_, the energy generated by the combination of atoms in the system, is an inherent property of the atom itself and an essential factor affecting the structure of the alloy phase. Ordinarily, the ∆H_mix_ in HEAs is either positive or negative. When the ∆H_mix_ is positive, the solid solubility of the alloy decreases with the increasing ∆H_mix_. When the ∆H_mix_ of the alloy system is negative, it is easy to generate intermetallic compounds. Therefore, when the absolute value of ∆H_mix_ is smaller, it is easier to form a solid solution [20]. The formula for calculating the mixing enthalpy generated when n different principal elements are randomly mixed is:(2)$${\Delta H}_{\mathrm{mix}}=\sum_{i=1,j\neq 1}^{n}\Omega_{\mathrm{ij}}c_{i}c_{j}\;\Omega_{\mathrm{ij}}=4\times {\Delta H}_{\mathrm{AB}}^{\mathrm{mix}}$$
where Ω_ij_ is the mixing enthalpy corresponding to two relative elements in the alloy system, and ${\Delta H}_{\mathrm{AB}}^{\mathrm{mix}}$ is the enthalpy of mixing between two elements in an alloy system calculated according to the Miedema model [21] (Table 3). 

Entropy is a parameter in thermodynamics that describes the state of matter, encompassing configurational entropy, vibrational entropy, magnetic dipole entropy, and electron randomness entropy [9,18]. The latter three make only a small contribution to the HEA system. Research on HEAs involves qualitative analysis, so configuration entropy is mainly used and expressed as:(3)$${\Delta S}_{\mathrm{conf}}=-R\left(1/n\ln1/n+1/n\ln1/n+\ldots +1/n\ln1/n\right)=R\ln n$$
where n is the number of chemical elements in the alloy, and R is the gas constant. It can be seen from the expression that the entropy increases with increasing number of principal elements. When it reaches a certain level, the growth rate slows down. Mixing entropy has an important effect on the phase structure of alloys, with higher entropy promoting solid-solution formation and inhibiting intermetallic compound formation in HEAs. Zhang et al. synthesized the effects of mixing enthalpy and entropy on the stability of the solid solution and proposed the Ω criterion [16,17], which can be expressed as: (4)$$\Omega =\frac{T_{m}{\Delta S}_{\mathrm{mix}}}{\left|{\Delta H}_{\mathrm{mix}}\right|}$$
where T_m_ is the average melting point of each alloy system, ∆H_mix_ is the enthalpy of mixing, and S_mix_ is the entropy of mixing. From thermodynamics, a solid-solution phase stability is related to its free energy difference before and after formation. For HEA, a lower ∆H_mix_ and higher temperature mean the easier formation of a stable HEA solid solution. When Ω > 1, the driving force is greater than the resistance and the probability of solid-solution formation is high. Considering theoretical deviation, Zhang et al. modified the Ω criterion and proposed that when Ω ≥ 1.1 and δ ≤ 6.6%, HEA solid solutions are more easily formed [16,17]. 

The ∆χ (electronegativity) and VEC (valence electron concentration) of the alloy also have an effect on the phase structure of the HEA and can be expressed as:(5)$$\Delta \;\chi =\sqrt{\sum_{i=1}^{n}c_{i}{\left(\chi_{i}-\;\bar{\chi }\right)}^{2}}\;\bar{\chi }=\sum_{i=1}^{n}C_{i}\chi_{i}$$
(6)$$\mathrm{VEC}=\sum_{i=1}^{n}c_{i}{\left(\mathrm{VEC}\right)}_{i}$$
where χ_i_ is the electronegativity and (VEC)_i_ is the valence electron concentration of the element. The ∆χ of alloying elements refers to the ability of elements to attract electrons in the alloy system. The greater the ∆χ difference, the easier the formation of intermetallic compounds. Therefore, to promote a solid solution and inhibit intermetallic compound formation, the smaller the ∆χ difference of the alloy system, the better. The VEC of the alloy system can predict the most likely phase structure of the solid solution. When the VEC changes or reaches a certain value, it will affect the stability of the bonds between the principal components and the stability of the solid solution. A higher VEC (≥8.6) is favorable for the formation of an FCC solid solution, a lower VEC (≤6.87) is favorable for the formation of a BCC solid solution, and an intermediate VEC (6.87 ≤ VEC ≤ 8.6) is favorable for the formation of a BCC+FCC dual-phase solid solution [22]. 

Table 4 shows the δ, ∆H_mix_, ∆S_mix_, Ω, ∆χ, and VEC values of the four alloys, summarized from experimental data and empirical criteria. The values are theoretical and should be verified empirically, since the experimental process and parameters can impact the results [23].

### 3.2. HEA Powders

MA is mainly achieved via ball milling, a process of mechanochemical synthesis in which the solid components are alloyed using mechanical energy [24]. The repeated collision and extrusion of the powder in the ball mill can cause repeated deformation, fracture, and welding so that it diffuses at the atomic level or produces a solid reaction, finally forming the alloy powder [25]. Therefore, the powder alloying degree, morphology, and microstructure largely depend on the process parameters of ball milling time and speed, the use of a process control agent (PCA), the ball/material ratio, milling ball size, ball milling mode and the milling atmosphere [26]. The effects of MA milling time, milling speed, and PCA on alloying behavior were studied for the Ni_35_Co_35_Cr_12.6_Al_7.5_Ti_5_Mo_1.68_W_1.39_Nb_0.95_Ta_0.47_ HEA powder system, which had considerable solid-solution capacity. 

#### 3.2.1. Effect of Milling Time

Ball milling time is a key parameter affecting powder phase structure and microstructure. Figure 2 gives the XRD pattern of the powder when the milling time is the only variable (up to 80 h), the milling speed is 300 r min^−1^, and the PCA is ethanol. The diffraction peak broadened as milling time increased, showing that the powder particle size steadily reduced, an effect attributed to crystallite refinement, decreased crystallinity, and aggregated lattice strain [27,28,29]. In the early stage of ball milling, diffraction peaks corresponding to each metal element are evident. After 50 h milling, the powder is completely alloyed and the dual phase BCC+FCC structure is formed; the FCC phase is the main phase. Until 80 h, the phase structure does not change, but the diffraction peak widens. Figure 3 gives SEM photographs of the powder at different milling time. The powder becomes a non-uniform block after milling for 10 h, mainly because, at the early stage of milling, initial spheroidal powders constantly collided with and are squeezed by balls of different sizes. After milling for 40 h, the phenomenon of powder agglomeration begins due to a cycling process of fracture into fine particles and the cold welding of the broken powder. After 60 h and 80 h, the powder agglomerates and stacks together to form a solid solution showing complete alloying, as indicated by the XRD pattern of Figure 2.

#### 3.2.2. Effect of Milling Speed

Similarly, the milling speed is another key MA parameter affecting the temperature in the spherical tank, the degree of powder refinement, and the degree of powder alloying. It is generally believed that the higher the ball milling speed, the greater the energy applied to the powder, facilitating alloying. Figure 4 shows the powder phase structure at different times when the milling speed is increased to 500 r min^−1^, the total time is 80 h, and the PCA is ethanol. After 30 h, the powder has been alloyed completely. The position of the diffraction peak has not changed, and a dual-phase BCC+FCC structure is formed (FCC is the main phase), indicating that increasing milling speed does not affect the alloying process. As shown in Figure 5, the powder particle size is smaller, and the agglomeration is closer, since increasing milling speed intensifies collision between the powder and the balls, and intensifies the cycling process of cold welding and crushing.

#### 3.2.3. Effect of PCA

A process control agent (PCA), which can be a solid, liquid or gas, is essential in MA to prevent excessive cold welding during ball milling, to prevent the powder from adhering to the milling balls and the inner wall of the milling tank, and to homogenize and improve the yield of the powder. Different PCAs can produce different results [30]. Generally, the loading of PCA is 1–5 wt% of the total powder. Excessive PCA not only pollutes the powder but also affects atomic diffusion and inhibits alloying [31]. Common PCAs include stearic acid, ethane, methanol, and ethanol. Figure 6 shows the phase structure with different milling times and stearic acid as the PCA. With the extension of milling time, the diffraction pattern does not change after 20 h, while the diffraction peak disappears after 30 h, and the powder is completely alloyed after 40 h, forming a dual-phase structure of BCC+FCC, in which FCC is the main phase. Compared with ethanol as PCA, the powder alloying time is extended from 30 to 40 h, indicating that when stearic acid is the PCA, the alloying process is inhibited to some extent, requiring a longer milling time. Figure 7 shows the SEM images of the powders when ethanol and stearic acid were used as PCA at a milling speed of 500 r min^−1^ for 30 h. As shown in Figure 7b, the powders were mostly long flakes with poor agglomeration since, in the early stage of ball milling, the powder is easily broken into fine and irregular particles under the collision and extrusion of the balls. The specific surface area and surface energy of the powder increase as the powder particle size decreases, resulting in an unstable energy state, and the consequent aggregation of the crushed powder particles to reduce the powder surface energy and form a more stable state [32]. However, when stearic acid is used as PCA, it adheres to the surface of the powder during the early stage of ball milling, inhibiting the aggregation of the powder particles, resulting in a crushed powder mainly comprised of tiny sheets. At the same time, because a layer of stearic acid envelops the powder, the powder alloying process will be slowed down. Until the stearic acid disperses after prolonged ball milling, the powder alloying process will not proceed. Figure 8 shows the curve of milling time and Dv50 under different milling parameters. When ethanol was PCA, the powder particle size decreased significantly after 10 h of ball milling, indicating that the powder was broken into fine particles under the impact and extrusion of the milling balls. However, when stearic acid was used as PCA, the powder particle size decreased less, further indicating that stearic acid adheres to the powder surface preventing its agglomeration and causing the powder to exist in the form of flakes. On stearic acid dispersion after 30 h, the powder particle size decreased significantly. On the whole, the powder particle size at 500 r min^−1^ is smaller than at 300 r min^−1^, indicating that the powder can be refined by increasing the ball milling speed.

In the MA process, it is necessary to strictly control the content of elemental O and C in the powder [33]. Figure 9 shows the content of O and C in the powder with time, revealing a slight increase with an increasing milling time, mainly due to the decomposition of the ethanol PCA. When stearic acid is used as PCA, the content of O and C is basically maintained in an equilibrium state and, although there are changes, the content of both is always less than 0.05 wt%. Experiments were carried out in an anaerobic argon environment, but it is very challenging to completely avoid contamination during mechanical alloying. As an inherent feature of the powder metallurgy process [34], possible contamination sources can only be controlled and mitigated.

To optimize experimental efficiency in the subsequent SPS step, the powder ball milling parameters selected were a milling time of 60 h, milling speed of 300 r min^−1^, and ethanol as PCA. To ensure the complete alloying of the powder, agglomerates in the powder after milling were analyzed using EDS. As shown in Figure 10, the elements are evenly distributed. Combined with the XRD results of Figure 2, the complete alloying of the powder is revealed under the selected milling parameters.

### 3.3. HEA Blocks

Spark plasma sintering (SPS) is a process that integrates heating and pressure, compressing the powder into dense blocks in a short time, and requiring a large quantity of powder. The alloy powder prepared with a milling speed of 300 r min^−1^ for 60 h and ethanol PCA was selected as the SPS raw material to prepare the HEA blocks. Microstructure, phase structure, and the properties of the HEA blocks sintered at different temperatures were studied. To determine the temperature range required for SPS, the alloyed powder was analyzed using DSC, with the results shown in Figure 11. The heat flux with temperature follows a trend of rising, falling, and rising again. The upward trend is endothermic, while the downward one is exothermic. Because the milling PCA was ethanol, which is volatile with a boiling point of 78 °C, the first endothermic peak at 76.9 °C corresponds to ethanol volatilization. With the increasing temperature, a second endothermic peak occurs at 276.3 °C. Given that the crushing and rolling of the milling ball forms powders with particle sizes of several hundred nanometers, the endothermic peak is caused by nanoparticle melting. At the same time, because lattice distortion of the powder occurs from prolonged milling ball collisions, the strain energy held in the powder can be released on heating. As a result, the first exothermic peak arises at 305.9 °C, and heat absorption and release happen simultaneously, resulting in a reasonably flat transition between 400 and 600 °C. Similarly, heat absorption induced by particle melting and heat release caused by internal energy release occur at around 700–900 °C. MA is a non-equilibrium process. The powder after ball milling will be in a metastable state, and a higher temperature will cause the powder to undergo exsolution and phase change. Therefore, phase transformation is likely to occur at 974.8 °C and 1075.4 °C. Five SPS sintering temperatures of 950, 1000, 1050, 1100, and 1150 °C were selected based on the DSC curve data, and the microstructure and phase structure of the block sintered at these temperatures were studied.

#### 3.3.1. Microstructure and Structure

XRD analysis of the microstructure and phase structure of the alloys after SPS is given in Figure 12. The phase structure changed at 950 °C, consistent with the prediction based on the DSC curve. The powder changed from a dual-phase to a single FCC phase structure, possibly due to a solid-solution transformation of BCC to FCC phase. With the increasing sintering temperature, a small amount of interstitial solid solution TiC is generated, possibly due to the interaction between the HEA powder and carbon in the SPS graphite mold, resulting in surface carburizing. Carbon atoms enter the lattice of HEA, forming carbides and causing lattice distortion, one of the four major characteristics of HEAs. As a basic structural parameter of crystal materials, the lattice parameter is typically used to determine the degree of lattice distortion [35]. According to Bragg’s law and the interplanar spacing of the cubic crystal system, the formula for calculating lattice parameters is as follows:(7)$$a=\frac{\lambda }{2\sin\theta }\sqrt{h^{2}+k^{2}+l^{2}}$$
where a is the lattice parameter, θ is the angle of diffraction, λ is the wavelength of the radiation source (λ = 0.15406 nm), and h, k, and l describe the indices of the θ crystal face.

The lattice parameters of MA and SPS samples are shown in Table 5. Before SPS, the powder lattice parameters of the BCC and FCC solid solutions, calculated from the positions of the (111) BCC and (110) FCC centers of gravity, are a = 0.3533 and 0.3163 nm, respectively. After SPS, the block lattice parameter of the FCC solid solution calculated from the position of the (111) FCC center of gravity is a = 0.4435 nm. This indicates that in the SPS process, FCC and BCC solid-solution phases are dissolved together, and the lattice constant becomes larger, forming a single FCC phase. Compared with the previous phase structure calculation, the phase structure after sintering is inconsistent with the predicted results, which further illustrates that the phase formation rules for HEA are empirical criteria, and that results may differ under different processes and parameters.

SEM images of the block after different sintering temperatures are shown in Figure 13. The density of the alloy block increases with the increasing sintering temperature and there are two contrasting microstructure regions: a grayish brown region A and a white region B. Region B is distributed around region A (Figure 13e). Component analysis of regions A and B (Table 6) indicates that region A’s composition is very close to the nominal composition, and that region B is mainly rich in Ti and C elements. Combined with XRD analysis results, region A is the FCC matrix phase, and region B is the interstitial solid solution TiC.

#### 3.3.2. Mechanical Properties

The mechanical properties of the HEA blocks sintered at different temperatures are shown in Table 7. The theoretical density of the HEA block is 8.022 g cm^−3^, and the real density ρ and relative density K increased with an increasing sintering temperature. At 1150 °C, the density reached a maximum of 7.92 g cm^−3^, with a relative density of 98.7%, demonstrating that the alloy’s densification is greatest at this temperature. In general, the atomic diffusion activation energy is relatively high, and the diffusion of atoms is difficult, requiring higher temperatures. Increasing sintering temperature will facilitate the combination of powder particles and reduce the number and size of pores, so the density of the alloy is positively correlated with the sintering temperature. The hardness of the alloy also increases with an increasing sintering temperature, with the maximum hardness reaching 1050 HV at 1150 °C, which is about twice the hardness of FGH100L (400–500 HV) under different processes.

The particular lattice distortion and retarded atomic diffusion of HEA may account for its high hardness [36]. Dislocation, as a microscopic imperfection, significantly impacts the mechanical characteristics of materials. Lattice deformation will impede dislocation migration and increase dislocation density, hence strengthening dislocations and enhancing alloy strength and hardness [37,38]. The slow HEA diffusion effect can hinder the formation of adverse effects such as grain coarsening and recrystallization at high temperatures, inhibit the diffusion of atoms, and improve the structural stability of the alloy [9,39]. The hardness values of the samples obtained at 950 and 1000 °C are quite different, mainly due to the unique lattice distortion effect of the HEA. The material has better sintering characteristics and lower porosity at 1000 °C and, with an increasing temperature, the hardness increases significantly.

The room temperature compressive stress–strain curves of HEA sintered at different temperatures are shown in Figure 14a. The figure shows no evident yield point, with high strength and good ductility. The compressive strength and strain steadily increase as the sintering temperature rises. The compressive strength is 2363 MPa, and the fracture strain is 18.3% at 1150 °C. To demonstrate the superior performance of the study HEA, Figure 14b depicts the fracture strength and strain of several other HEAs prepared in the same manner [27,40,41,42,43,44,45,46,47,48]. When compared to other HEAs, the alloys in this experiment not only have higher strength, but also have higher strain. In general, strength and ductility are inversely proportional, with ductility decreasing as strength increases. As a result, it is vital and valuable to investigate the synergistic mechanism of strength augmentation and ductility improvement. In general, BCC solid solutions have higher strength and hardness but poor ductility, whereas FCC solid solutions have superior ductility but lower strength [9]. However, in this experiment, the FCC solid solution had both good ductility and high strength. Solid-solution strengthening, as one of the four metal material strengthening procedures, is regarded as an efficient approach to increasing HEA strength [49,50]. The HEA lattice distortion effect increases resistance to dislocation movement, making slip difficult, and improving the strength of the alloy solid solution. In summary, the strengthening mechanism of the alloy with higher hardness, strength, and larger strain is mainly solid-solution strengthening caused by lattice distortion. The good general compressive properties of the alloy indicate a wide applicability in practical engineering. The microstructure at the fracture of the HEA compression specimen studied (Figure 15) reveals a large number of fluvial patterns and dissociation steps caused by stress concentration at the fracture, and many small cleavage steps formed by dislocation movement, which then converge to form a fluvial pattern. The fracture form is a typical cleavage fracture of the brittle fracture type, consistent with the results of the compression stress–strain curve.

## 4. Conclusions

Based on FGH100L, Ni_35_Co_35_Cr_12.6_Al_7.5_Ti_5_Mo_1.68_W_1.39_Nb_0.95_Ta_0.47_ HEA was designed according to the composition design and the phase formation rules for HEAs. The preparation of the study HEA powder and block using the MA-SPS method provides a new avenue for the design and preparation of a new generation of powder superalloy turbine disks. The optimum ball milling parameters and sintering temperature, the phase structure and microstructure of the powder and block, and the alloy properties were investigated with the following results.

In the Ni_35_Co_35_Cr_12.6_Al_7.5_Ti_5_Mo_1.68_W_1.39_Nb_0.95_Ta_0.47_ HEA alloy system, δ, ΔH_mix_, ΔS_mix_, Ω and ∆χ are all within the standard range of the HEA solid-solution formation rules. The VEC is 8.0898, with a predicted BCC+FCC dual-phase structure.The HEA system powder was successfully synthesized using the MA method, which, according to the predicted results, formed the BCC+FCC dual-phase structure. The powder can be completely alloyed at a milling speed of 300 r min^−1^ for 50 h, with ethanol as PCA. Powder particle size can be reduced by increasing the milling speed. Milling time and speed have no effect on the powder alloying process. With stearic acid as PCA, the alloying process was hindered, requiring increased milling time to complete the powder alloying.The HEA block was successfully prepared using SPS, and the phase structure changed from dual-phase to a single FCC phase at 950 °C. The density and hardness of the HEA block are positively correlated with temperature. The main strengthening mechanism is solid-solution strengthening. At 1150 °C, the density of the alloy is 7.92 g cm^−3^, the relative density is 98.7%, and the maximum hardness is 1050 HV. The maximum compressive strength of the alloy is 2363 MPa, with no evident yield point and fluvial patterns and gradation steps at the fracture. The fracture mode is of the typical cleavage type, a brittle fracture type.

## Figures and Tables

**Figure 1 materials-16-02082-f001:**
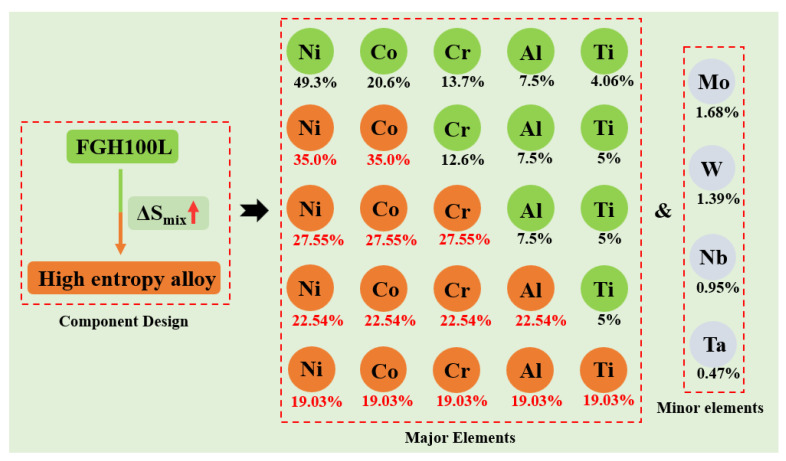
High entropy design based on FGH100L.

**Figure 2 materials-16-02082-f002:**
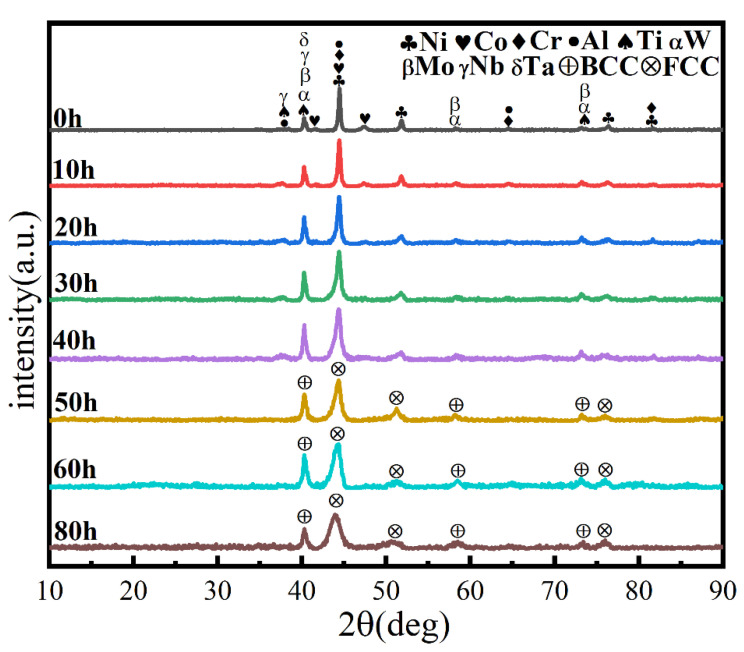
XRD patterns of powders at different milling times with a milling speed of 300 r min^−1^ and ethanol PCA.

**Figure 3 materials-16-02082-f003:**
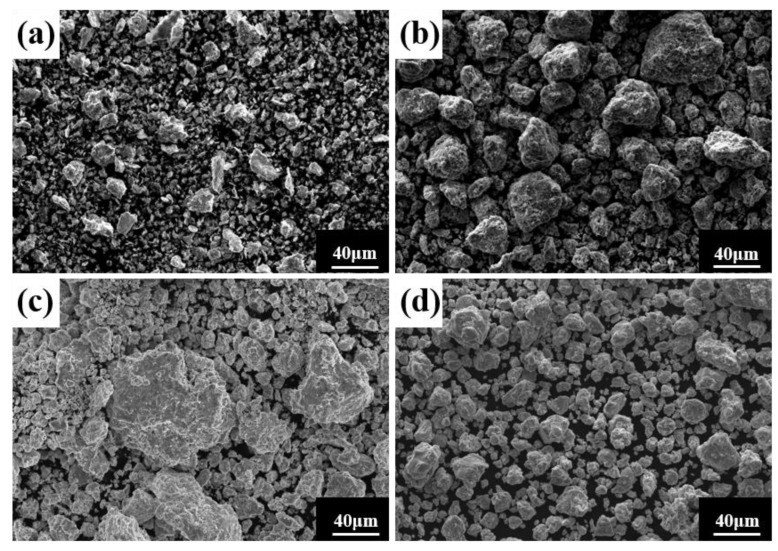
Powder microstructures at different milling times with a milling speed of 300 r min^−1^ and ethanol PCA: (**a**) 10 h; (**b**) 40 h; (**c**) 60 h; and (**d**) 80 h.

**Figure 4 materials-16-02082-f004:**
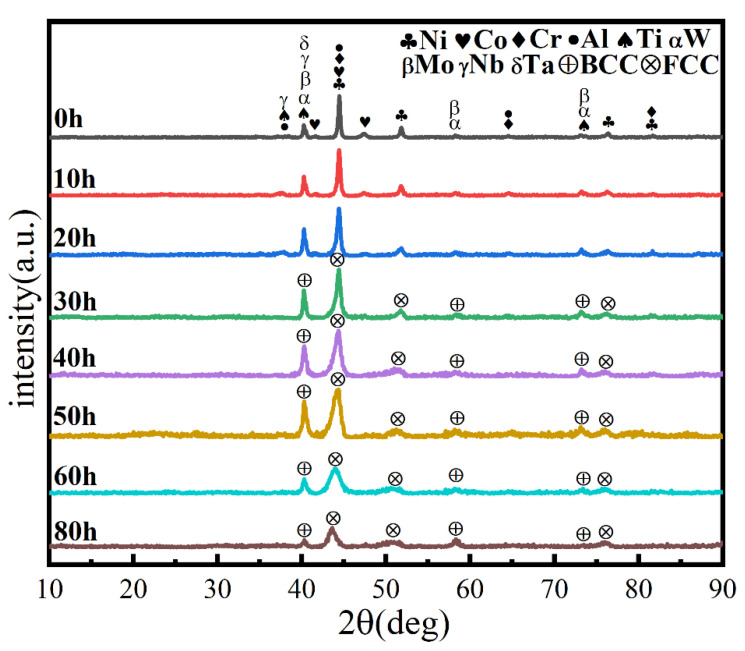
XRD patterns of powders at different milling times with a milling speed of 500 r min^−1^ and ethanol PCA.

**Figure 5 materials-16-02082-f005:**
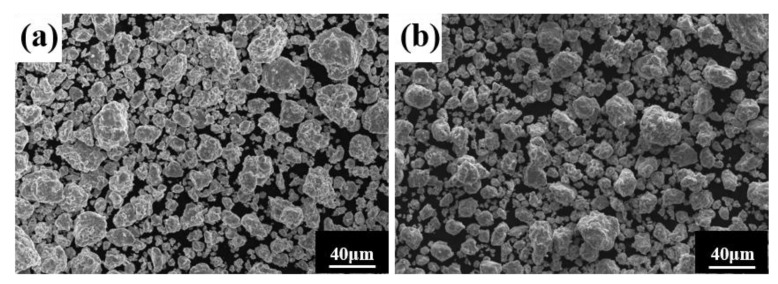
Powder microstructures at different milling times with the a milling speed of 500 r min^−1^ and ethanol PCA: (**a**) 30 h; and (**b**) 60 h.

**Figure 6 materials-16-02082-f006:**
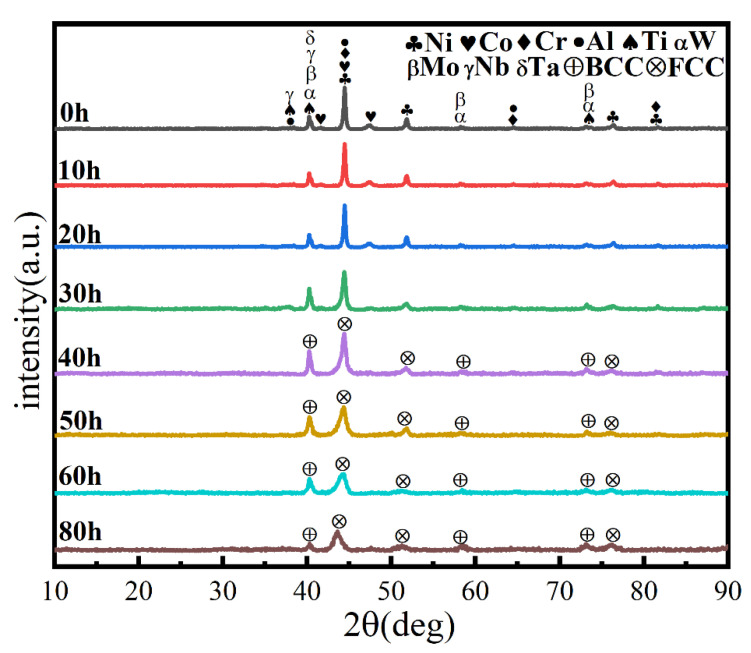
XRD patterns of powders at different milling times at a milling speed of 500 r min^−1^ and with stearic acid as PCA.

**Figure 7 materials-16-02082-f007:**
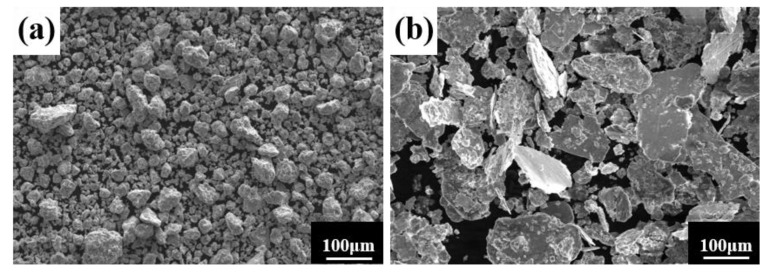
Powder microstructures with different PCA at a milling speed of 500 r min^−1^ for 30 h: (**a**) ethanol; and (**b**) stearic acid.

**Figure 8 materials-16-02082-f008:**
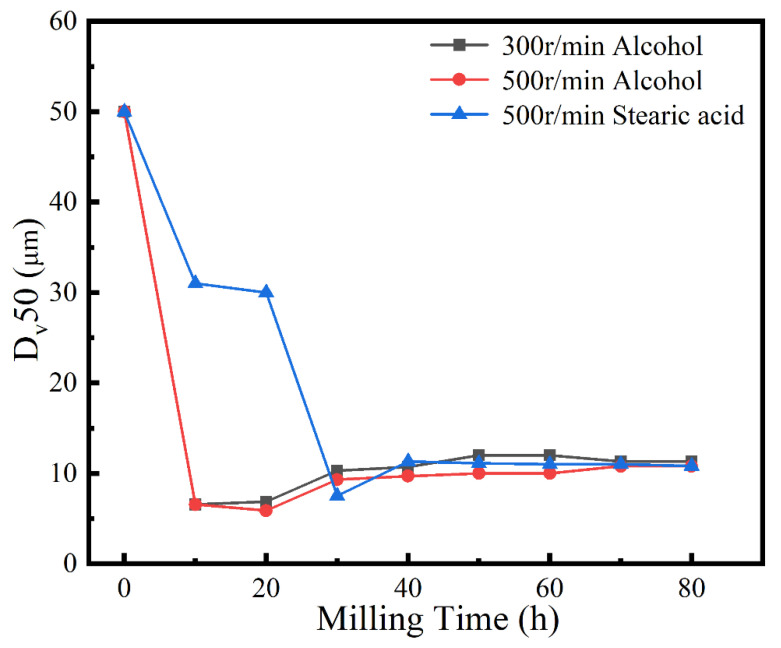
Powder D_v_50 under different milling parameters.

**Figure 9 materials-16-02082-f009:**
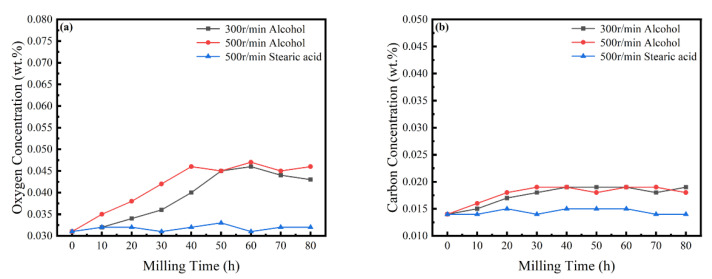
Powder O and C concentrations at different milling times. (**a**) O concentrations; and (**b**) C concentrations.

**Figure 10 materials-16-02082-f010:**
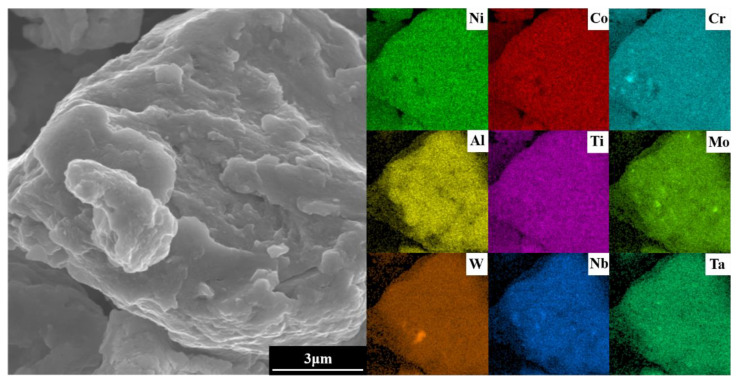
EDS images of the agglomerates in the alloy powder.

**Figure 11 materials-16-02082-f011:**
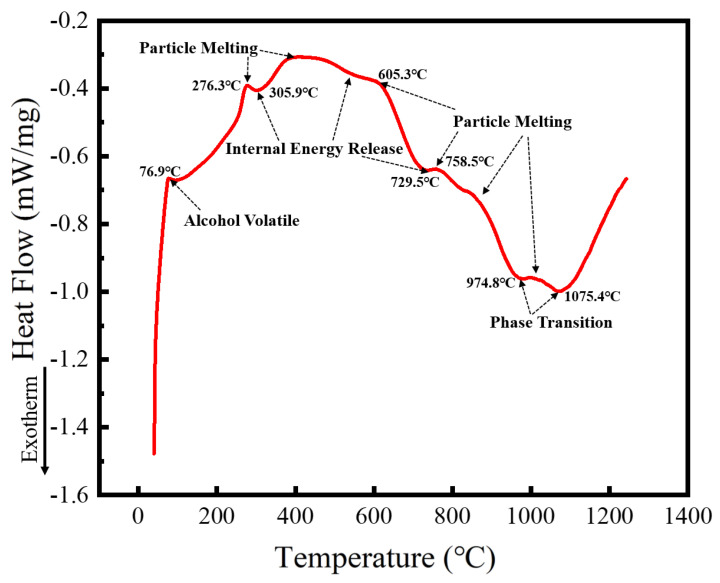
DSC curves of powders after MA.

**Figure 12 materials-16-02082-f012:**
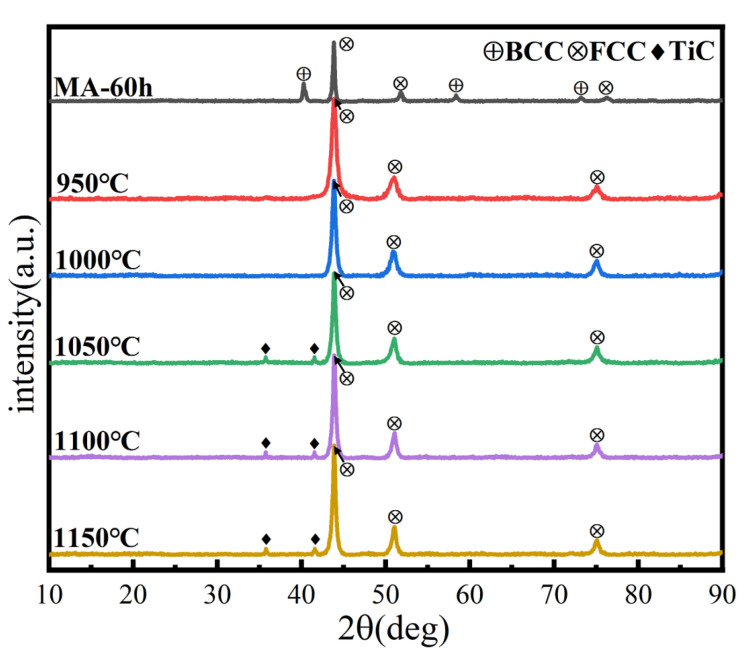
XRD patterns of the HEA blocks at different sintering temperatures.

**Figure 13 materials-16-02082-f013:**
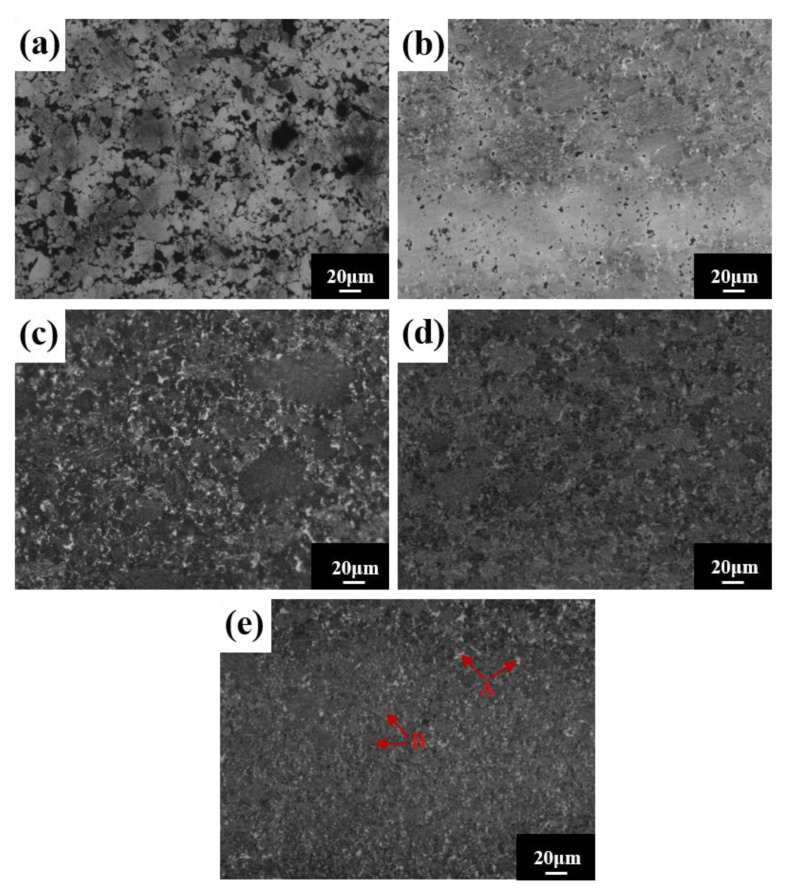
Microstructures for the HEA blocks at different sintering temperatures: (**a**) 950 °C; (**b**) 1000 °C; (**c**) 1050 °C; (**d**) 1100 °C; and (**e**) 1150 °C.

**Figure 14 materials-16-02082-f014:**
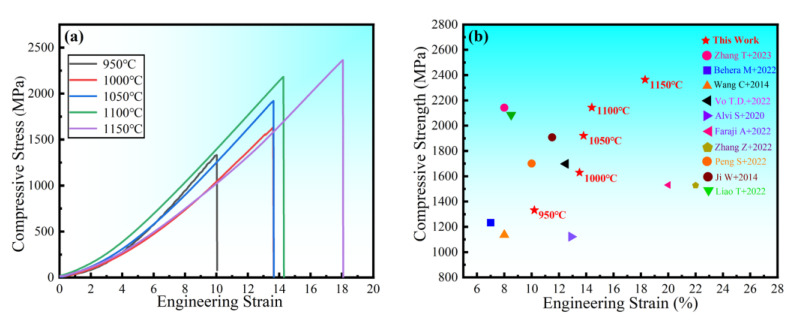
Engineering stress–strain curve of the block under compression (**a**) at room temperature and (**b**) strength compared with other HEAs [27,38,39,40,41,42,43,44,45,46].

**Figure 15 materials-16-02082-f015:**
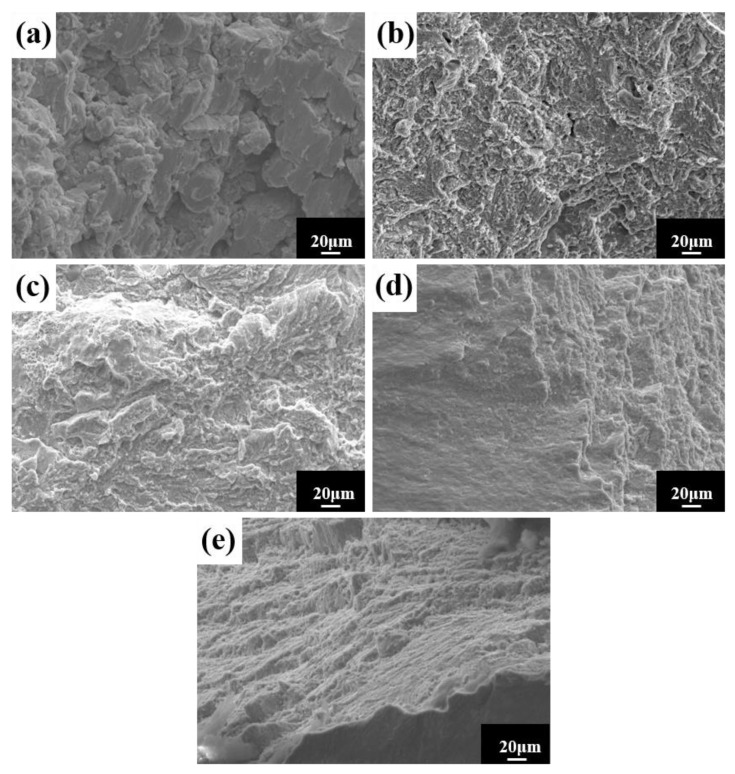
Compression fracture morphology of HEA at different sintering temperatures: (**a**) 950 °C; (**b**) 1000 °C; (**c**) 1050 °C; (**d**) 1100 °C; and (**e**) 1150 °C.

**Table 1 materials-16-02082-t001:** Nominal composition of powder superalloy FGH100L (at.%).

Elements	Ni	Co	Cr	Al	Ti	Mo	W	Nb	Ta	C+B+Zr
at %	49.3	20.6	13.7	7.5	4.06	1.68	1.39	0.95	0.47	little

**Table 2 materials-16-02082-t002:** Characteristic parameters of alloying elements.

Element	Structure	Molar Mass (g mol^−1^)	Average Atomic Radius (nm)	Melting Point (°C)	VEC (e a^−1^)	Electronegativity	Density (g cm^−3^)
Ni	FCC	58.69	0.1240	1455	10	1.91	8.9
Co	FCC/HCP	58.93	0.1251	1495	9	1.88	8.9
Cr	BCC	52.00	0.1249	1850	6	1.66	7.2
Al	FCC	26.98	0.1432	660	3	1.61	2.7
Ti	HCP	47.86	0.1432	1660	4	1.54	4.5
Mo	BCC	95.94	0.1363	2625	6	2.16	10.2
W	BCC	183.84	0.1367	3407	6	2.36	19.35
Nb	BCC	92.91	0.1429	2415	5	1.60	8.57
Ta	BCC	180.94	0.1430	2996	5	1.50	16.68

**Table 3 materials-16-02082-t003:** Mixing enthalpy of binary atomic pairs in various alloy systems (kJ mol^−1^) [21].

Element	Co	Cr	Al	Ti	Mo	W	Nb	Ta
Ni	0	−7	−22	−35	−7	−3	−30	−29
Co		−4	−19	−28	−5	−1	−25	−24
Cr			−10	−7	0	1	−7	−7
Al				−30	−5	−2	−18	−19
Ti					−4	−6	2	1
Mo						0	−6	−5
W							−8	−7
Nb								0

**Table 4 materials-16-02082-t004:** The calculated values of δ, ∆Hmix, ∆Smix, Ω, ∆χ, and VEC for the HEA.

Alloy System	δ	ΔH_mix_	ΔS_mix_	Ω	∆χ	VEC
FGH100L	5.2723	−13.14	1.46R	1.643	0.1463	8.2486
Ni_35_Co_35_Cr_12.6_Al_7.5_Ti_5_ Mo_1.68_W_1.39_Nb_0.95_Ta_0.47_	5.4159	−13.03	1.53R	1.730	0.1462	8.0898
Ni_27.55_Co_27.55_Cr_27.55_Al_7.5_Ti_5_ Mo_1.68_W_1.39_Nb_0.95_Ta_0.47_	5.4126	−13.24	1.60R	1.813	0.1548	7.5685
Ni_22.54_Co_22.54_Cr_22.54_Al_22.54_Ti_5_ Mo_1.68_W_1.39_Nb_0.95_Ta_0.47_	6.5883	−18.48	1.69R	1.304	0.1622	6.7664
Ni_19.03_Co_19.03_Cr_19.03_Al_19.03_Ti_19.03_ Mo_1.68_W_1.39_Nb_0.95_Ta_0.47_	7.2631	−24.71	1.77R	1.049	0.1664	6.3454

**Table 5 materials-16-02082-t005:** Lattice parameters of Ni_35_Co_35_Cr_12.6_Al_7.5_Ti_5_Mo_1.68_W_1.39_Nb_0.95_Ta_0.47_ HEA during MA and SPS.

	Crystal Structure	Lattice Parameter (nm)	
		MA	SPS
Ni_35_Co_35_Cr_12.6_Al_7.5_Ti_5_ Mo_1.68_W_1.39_Nb_0.95_Ta_0.47_	BCC	0.3533	-
	FCC	0.3163	0.4435

**Table 6 materials-16-02082-t006:** EDS analysis of the HEA blocks at different regions (at. %).

Alloy Regions	Ni	Co	Cr	Al	Ti	Mo	W	Nb	Ta	C
Nominal	35	35	12.6	7.5	5	1.68	1.39	0.95	0.47	-
A	34.45	33.95	13.55	7.35	6.21	1.65	1.41	0.95	0.48	-
B	6.13	5.35	4.21	4.22	41.53	1.83	0.98	1.01	0.71	33.85

**Table 7 materials-16-02082-t007:** The mechanical properties of HEA blocks sintered at different temperatures.

Properties/T (°C)	950	1000	1050	1100	1150
ρ (g cm^−3^)	6.92	7.22	7.57	7.72	7.92
K (%)	86.30	90.10	94.40	96.30	98.70
Hardness (HV)	560	819	1018	1031	1050
Fracture strength (MPa)	1331	1628	1919	2143	2363
Compressive strain (%)	10.2	13.5	13.8	14.4	18.3

## Data Availability

The datasets generated and analyzed during the current study are available from the corresponding author on reasonable request.

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
