# Peer review of "Microstructure and Properties of NiCoCrAlTi High Entropy Alloy Prepared Using MA-SPS Technique"

_materials, 2023, doi:10.3390/ma16052082_

Round 1

Reviewer 1 Report

Dear Authors, 

Thank you for the nice work. This work explored the microstructure and mechanical property of the high entropy alloy synthesized by various milling time and sintering temperatures. I respect the technical rigor taken throughout the research effort.

The data look great, I can imagine your team have spent so much time performing a number of experiments. I do believe the report has a technical value to be published by the journal Materials

However, please consider the following comments, and hopefully take this as a chance to significantly increase the merit of your work. 

1. The x-ray data shown on multiple figures need to be better visualized. It is hard to find the peaks, you can freely increase the size of the figure to show the peaks clearly if needed. In Fig. 10, around 61 degrees, there are small peaks as high as those of TiC. Please review where does it come from. 

2. It is a pity to not having a transmission electron microscopy on this exciting material. X-ray diffraction may capture the sample scale microstructure, but if you want to claim that you successfully synthesized a HEA, it should be able to demonstrate a full crystallinity at a local level. In addition, local scale energy dispersive spectrum through TEM should be able to tell that your material is indeed a homogeneous HEA.

3. Recrystallization property of HEA is missing. We have data for the structure of materials whose sintering time lasted only for 2 minutes. Are we confident this material will stay as a stable HEA for long enough to call it a stable phase? It is desirable to test an additional sintering test to understand their recrystallization behavior. Without this information, it is equally possible that we are merely looking at a metastable transitioning phase of this material.

4. Unfortunate TiC formation. Do you perhaps regret using the graphite tool for your SPS process? Titanium rapidly reacted with the graphite to form TiC, and I don't think this was an intentional design of the experiment. I do believe there should have been a way to avoid this artifact.

5. Captions of figures need to expand. Please add more context and idea, some take-home message for the captions of each figures. It really helps reader to grasp the idea around the shown graphs if you provide more into the captions.

With these comments, I really hope to see a revision of this nice work. I was able to see how much details that you took care of during the research activity.

Though I recommend a major revision for this, but please take that as a chance to seriously review if this work can be substantially transform. I like the data as its current form, but I do believe there are more you can do to make it much bigger. Please consider the listed concerns that I raised above, and come back with a set of strong answers. I look forward to reading this again!

Best regards,

Peer reviewer

Reviewer 2 Report

This is a detailed and comprehensive article about the synthesis and investigation of the Microstructure and Properties of NiCoCrAlTi High Entropy Alloy Prepared by MA-SPS Technique. In this study, the authors have successfully synthesized and conducted detailed investigations to verify the characteristics of this HEA alloy and the influence of the MA and SPS process parameters. However, there are still parts that require correction and clarification, which can be summarized as follows:

1. Abstract:

Line 10 “The phase formation law of the alloy system was predicted.”, should add: “…predicted and examined by experiments.” Or the authors can omit this sentence because lines 14-15 repeat it again.

Line 16, “…increasing the milling speed refines the powder.”, is incorrect, the author should consider correcting it to “… increasing the milling speed leads to the reducing of powder size.”

Line 20, “at 1150 oC,” is the temperature of what process? Since this is a summary, it should be brief, but not lacking in the basic information.

2. Introduction:

If the authors have discussed numerous aerospace industry challenges, why don't the authors relate the results to these applications?

3. Experimental procedure:

About SPS, heating rate should be added.

The amount of alcohol and stearic acid should be added. 

4. Results and discussion:

The author has explained reasonably and fully the theory of this process as well as related calculations to design this HAE alloy system (section 3.1). However, there are still things that need to be clarified in the later experimental part:

Formula 7 should be correct. 

Figure 7 and the explanation from lines 281-290 are not clear, because the authors have not clarified the SEM image of the powder sample, in stearic acid, how long has been milled? If we prolong the milling time (80h), will the powder form still flake?

And, in the explanation from lines 288-292: “when stearic acid is used as PCA, it will adhere to the surface of the powder, … the process of powdering will be down.” What sources or studies did the authors rely on for this explanation?

Figure 9: The exotherm direction should be added. The discussion of DSC should be re-written for more clear.

In section 3.3.2 and table 7, the authors use the phrase “at different sintering temperatures” to refer to samples after performing SPS. However, the question is, did the authors perform the strength measurement at that temperature or after the sample had cooled? if the authors do it when it has cooled down, consider changing it to “of samples which are sintered at different temperatures” to make the explanation clearer.

5. Conclusion:

As mentioned above, the author needs to relate the applications of this alloy to the results of the paper, because the article only concludes the results achieved. So how effective are these results when applied to practice (turbine disk)?

In conclusion, this article provides the reader with a variety of results and explanations for them, but with some unclear aspects. The authors should consider expanding and revising the abstract, introduction, and conclusion to better suit the aerospace industry application.

Reviewer 3 Report

Dear authors,

kindly check the line no 14 - . It was discovered that the HEA phase>.

Line no 118-121 not required. Discuss only the salient findings with reference. Eqn can be refered as (i)

Fig 1 caption may be improved. What is the reason to do the effect of milling time. it is well known that increase the time leads to decrease the partcile size which enhances the densification. After milling did the authors did the D10 measurements? In general the speed may not impact on properties, however there may be a change in particle size may occur. Did the author compare those parameters in tabular format?

The density must be represented in relative to identify that how much densification occur on the samples. The test details and discussion must be improved for compression.

Reviewer 4 Report

This work is devoted to the study of the Ni35Co35Cr12.6Al7.5Ti5Mo1.68W1.39Nb0.95Ta0.47 high-entropy alloy. The presented work is of high quality, the degree of presentation of the material leaves no questions: a comprehensive study was performed, supported by theory. However, some questions need to be answered:

Please use in the manuscript “Figure” instead of “Fig.”

3. Results and discussions

            Figures 3, Figure 5, Figure 7. Can the authors provide results of EDS analysis of agglomerates in the mechanically-milled powders to estimate their elemental composition and compare the results with XRD analysis?

            Section 3.3. Lines 319-321 “and then the second endothermic peak appeared at 276.3 °C with the increase of temperature, which was caused by the melting endothermic of some powders” – Since your alloy contains elements with a high melting point and even refractory elements, this statement must be supplemented as to why particles melt at such a low temperature.

Section 3.3.1. Why the MA-60h powder was sintered, but not MA-80h?

            Section 3.3.2. Mechanical properties.

            Lines 388-391. “The high hardness of HEA may be caused 388 to its unique lattice distortion and sluggish diffusion effect. The lattice distortion will im- 389 pede dislocation migration and raise the alloy's hardness. The slow diffusion effect can 390 prevent atom diffusion in the alloy and increase its structural stability[9].This statement should be expended and supplemented by other works. The large difference in hardness values between samples obtained at 950 and 1000 °C can be explained by better sinterability and lower porosity of the material at 1000 °C. A further increase in hardness with increasing temperature can be due to the formation of titanium carbide particles with a hardness above 28 GPa.

Round 2

Reviewer 1 Report

Dear Authors, 

Thank you for your hard work. I am impressed by the revised manuscript.

There are some weaknesses of the current paper, however, I believe the current form of the manuscript must be useful for many in this field. 

Here are the remaining criticism on the paper. 

In your conclusion #2, please revise this line "Increasing the milling speed to 500 r min-1 can not only shorten the alloying time but also refine the powder."  The double negative phrase is confusing. Please rewrite this line to a more direct, succinct expression.

In your conclusion #3, you are claiming that the TiC is an important strengthening phase "The TiC interstitial phase was formed on the basis of FCC main phase at 1050 °C, as an important strengthening phase".

But you have discussed that the TiC phase was formed unintentionally. "a small amount of interstitial solid solution TiC is generated, which may be due to the interaction between HEA powder and carbon in graphite grinding tools

I don't really think you can claim the benefit of TiC phase that was introduced out of control, as a conclusion of your experiment. This makes your graphite grinding tool one of a key ingredient of your HEA, which doesn't make sense. 

The same criticism goes to the #3 of the Conclusion. I believe it is a mistake to claim the benefit of having TiC, when it was formed unintentionally by using graphite apparatus. You could have chosen an inert grinding tool, and your work was not able to control the amount of carbon that was introduced, so it does not yield a useful knowledge along the line of thought.

If you really need to claim the benefit of TiC formation, probably this work needs to add much more data related to how much carbon have you introduced to the alloy batch.

You have enough data to conclude around the syntheses of HEA, even without highlighting the mechanical strengthening effect of TiC. Your technical observation, experimental tests on milling time and solvents are highly valuable technical information. I believe your conclusion can stand in line with those "controlled" variables and their impacts.

Other problems that I pointed out earlier, have been nicely addressed. I like how you have significantly improved the XRD plots! Nice work!

Unfortunately I have to recommend a major revision since the conclusion has gone out of consistency. Please revise the key message of your work. I respect your hard work, and I can image so much time that your team spent on the experimental data. Please focus on the real value of your data and draw out consistent conclusions with your control variables. (You can still discuss on the strengthening effect of TiC and its unintentional formation.)

(If you mean to publish on the coincidental formation of TiC and its benefit, the title of your paper probably has to change, and most likely a materials scientist like me will ask you to show a transmission electron microscopy around the TiC phase in your matrix.)

Reviewer 2 Report

The manuscript has undergone a significant revision.

The following are some of the things that need to be changed: Figure 10 should have the correct unit of heat flow, which is microwatts per milligram of mass.

Reviewer 4 Report

The article can be published in the current form
